# Transcriptome Profile Reveals Genetic and Metabolic Mechanisms Related to Essential Fatty Acid Content of Intramuscular *Longissimus thoracis* in Nellore Cattle

**DOI:** 10.3390/metabo12050471

**Published:** 2022-05-23

**Authors:** Gustavo Pimenta Schettini, Elisa Peripolli, Pâmela Almeida Alexandre, Wellington Bizarria dos Santos, Angélica Simone Cravo Pereira, Lúcia Galvão de Albuquerque, Fernando Baldi, Rogério Abdallah Curi

**Affiliations:** 1School of Agricultural and Veterinary Sciences, São Paulo State University, Jaboticabal 14884-900, SP, Brazil; wellington.bizarria@unesp.br (W.B.d.S.); galvao.albuquerque@unesp.br (L.G.d.A.); fernandobaldiuy@gmail.com (F.B.); 2School of Veterinary Medicine and Animal Science, University of São Paulo, Pirassununga 13635-900, SP, Brazil; elisa_peripolli@hotmail.com (E.P.); angelpereira@usp.br (A.S.C.P.); 3Commonwealth Scientific and Industrial Research Organization, Agriculture & Food, St Lucia, QLD 4067, Australia; pama.alexandre@gmail.com; 4School of Veterinary Medicine and Animal Science, São Paulo State University, Botucatu 18618-681, SP, Brazil; rogerio.curi@unesp.br

**Keywords:** *Bos taurus indicus*, differential co-expression, RIF, DH, PCIT, WGCNA

## Abstract

Beef is a source of essential fatty acids (EFA), linoleic (LA) and alpha-linolenic (ALA) acids, which protect against inflammatory and cardiovascular diseases in humans. However, the intramuscular EFA profile in cattle is a complex and polygenic trait. Thus, this study aimed to identify potential regulatory genes of the essential fatty acid profile in *Longissimus thoracis* of Nellore cattle finished in feedlot. Forty-four young bulls clustered in four groups of fifteen animals with extreme values for each FA were evaluated through differentially expressed genes (DEG) analysis and two co-expression methodologies (WGCNA and PCIT). We highlight the *ECHS1*, *IVD*, *ASB5*, and *ERLIN1* genes and the TF *NFIA*, indicated in both FA. Moreover, we associate the *NFYA*, *NFYB*, *PPARG*, *FASN*, and *FADS2* genes with LA, and the *RORA* and *ELOVL5* genes with ALA. Furthermore, the functional enrichment analysis points out several terms related to FA metabolism. These findings contribute to our understanding of the genetic mechanisms underlying the beef EFA profile in Nellore cattle finished in feedlot.

## 1. Introduction

Beef is a source of fats, proteins, vitamins, and minerals necessary in several metabolic pathways in humans [1]. Among the fatty acids (FA), the basic unit of lipids, the beef provides essential fatty acids (EFAs) such as linoleic (LA—C18:2 n6) and alpha-linolenic (ALA—C18:3 n3). These FA are not naturally synthesized by mammals and have been associated with lower incidences of inflammatory and cardiovascular diseases in humans [2]. The lipid absorption and deposition in cattle are influenced by the effects of genetic and ruminal metabolism [3,4,5,6].

Moreover, based on genetic parameters and variability, the FA profile, especially LA and ALA, can improve their concentration in beef through genomic selection [7,8]. This is an important feature since the FA profile is a difficult and expensive trait to measure.

Although genomic-wide association studies (GWAS) have provided information to support the understanding of complex traits, i.e., FA profile and intramuscular fat (IMF) deposition in cattle [9,10,11], this approach has displayed some limitations in pointing out genomic regions that explain a high proportion of genetic variance, as well as the cause-and-effect relationship among the single-nucleotide polymorphism (SNP) genotypes and EFA profile. This can be explained due to the complex and polygenic features of the FA profile. To address these limitations, the results generated by transcriptomic analyses such as differentially expressed genes (DEG), co-expression (COE), and differential co-expression (DCO) could be used.

Notwithstanding that the DEG analysis has been widely used to achieve a better understanding of genetic mechanisms, it is not sufficient to fully identify the gene groups related to complex traits since it does not allow the identification of potential genes that are not differentially expressed [12]. Hence, the DCO emerges due to the capacity to evaluate the global gene expression and its interrelationships to indicate sub-networks and better-connected genes to the phenotype [12,13,14].

The combined use of two co-expression algorithms, weighted gene co-expression network analysis (WGCNA) and partial correlation information theory (PCIT), can point out potential gene groups related to complex traits through different ways [15,16]. The WGCNA clusters genes in co-expression modules based on gene expression patterns between the samples, indicating gene groups correlated to the trait evaluated [15]. Otherwise, the PCIT incorporates theoretical information about transcription factors, which allows, through different metrics, the indication not only of hub genes but also the identification of transcription factors responsible for modulating the expression of differentially expressed genes between contrasting groups [12]. Moreover, the DCO can also assist the GWAS to indicate genomic regions with higher accuracy in integrated -omics studies [17,18].

Nevertheless, studies evaluating the beef FA profile of EFAs in indicine cattle (*Bos taurus indicus*) are incipient, especially incorporating different methodologies of differential co-expression analysis. Therefore, we aimed to identify potential regulatory transcription factors and hub gene groups related to the EFA profile in the Longissimus thoracis muscle of Nellore cattle finished in feedlot based on two methodologies, WGCNA and PCIT.

## 2. Results

### 2.1. Differentially Expressed Genes Analysis

The DEG were identified among the contrasting groups for LA and ALA. The LA groups encompassed 151 genes differentially expressed and enriched in 36 GO:Terms (2 BP, 2 MF, 34 CC) and six KEGG pathways, while the ALA groups displayed 352 DEG in 207 GO:Terms (102 BP, 18 MF, 87 CC) and 10 KEGG pathways (Appendix A). Among the genes identified, we can highlight the acyl-CoA oxidase 2 (*ACOX2*) and 3-hydroxybutyrate dehydrogenase type 1 (*BDH1*), listed as upregulated in the LA-H group and enriched in the terms “oxidation-reduction process” (GO:0055114) and “metabolic pathways” (bta01100). The genes succinate dehydrogenase complex iron sulfur subunit B (*SDHB*), enriched in the terms “oxidation-reduction process” (GO:0055114) and “oxidative phosphorylation” (bta00190), and the fat storage inducing transmembrane protein 1 (*FITM1*), inserted in the “metabolic process” (GO:0008152), were found upregulated in the ALA-H group. Moreover, the gene acyl-CoA synthetase long-chain family member 3 (*ACSL3*), also enriched in the GO:0008152, was found downregulated in the ALA-H group. Furthermore, the enoyl-CoA hydratase, short-chain 1 (*ECHS1*), and isovaleryl-CoA dehydrogenase (*IVD*) genes, involved in “oxidation-reduction processes” (GO:0055114) and “metabolic pathways” (bta01100), were upregulated in both groups (LA-H and ALA-H).

### 2.2. Differential Co-Expression Analysis (PCIT—DH)

The correlation matrices for each FA group were performed using the PCIT algorithm and only the correlations ≥|0.9| were used for the identification of DH genes between the contrasting groups. A total of 5649 genes inserted in 732 GO:Terms (419 BP, 115 MF, 198 CC) and 68 KEGG pathways were identified between LA-H and LA-L groups (Supplementary Files S3 and S4). For the ALA-H and ALA-L, a total of 5559 differential hubbing genes and 774 GO terms (454 BP, 106 MF, 214 CC) and 73 KEGG pathways were described (Appendix A). Of these, we can underscore the genes WD repeat domain (WDR43)—“regulation of catalytic activity” (GO:0050790) and “RNA binding” (GO:0003723); ankyrin repeat and SOCS box containing 5 (*ASB5*)—“protein ubiquitination” (GO:0016567); ER lipid raft associated 1 (*ERLIN1*)—“regulation of gene expression” (GO:0010468); and TRAF-type zinc finger domain containing 1 (*TRAFD1*), which presented a high number of connections in the LA-H and ALA-H groups (Appendix A and Table 1). However, when considering the low concentration groups, no common genes were found between LA and ALA.

### 2.3. Differential Co-Expression Analysis (PCIT—RIF)

In the RIF analysis, 20 potential TF were listed for each FA (Appendix A). Such TF were enriched in 86 GO:Terms (52 BP, 28 MF, 6 CC) to LA and 81 GO:Terms (38 BP, 31 MF, 12 CC) to ALA; in none of the fatty acids were there enriched KEGG pathways (Appendix A). Six TF were common to both fatty acids, in which we can highlight the nuclear factor IA (*NFIA)* and histone H4 transcription factor (*HINFP*), both enriched in the term “regulation of gene expression” (GO:0010468), as well as the zinc finger protein 473 (*ZNF473*), which was ranked with the more expressive RIF scores (Table 2 and Table 3).

More specifically to the LA, the enriched TF nuclear transcription factor Y subunit alpha (*NFYA*) and beta (*NFYB*)—“gene expression” (GO:0010467), as well as the BTB domain and CNC homolog 1 (*BACH1*) and nuclear factor of activated T cells 5 (*NFAT5*)—“regulation of biological process” (GO:0050789) positively impacted the target genes’ (DEG) expression in the LA-H group (Appendix A; Table 2). The presence of the TF zinc finger protein 134 (*ZNF134),* zinc finger protein 584 (*ZNF584*), zinc finger and BTB domain containing 43 (*ZBTB43*), and SLC2A4 regulator (*SLC2A4RG*) was also pointed out in the RIF analysis (Table 2). On the other hand, impacting more the DEG in LA-L, we highlight the TF WIZ zinc finger (*WIZ*)—“regulation of biological process” (GO:0050789) (Appendix A; Table 2).

With the emphasis on the ALA, the TF RAR-related orphan receptor A (*RORA*), nuclear transcription factor X-box binding 1 (*NFX1*), melanocyte inducing transcription factor (*MITF*), and activating transcription factor 6 (*ATF6*)—“gene expression” (GO:0010467)—and HIVEP zinc finger 2 (*HIVEP2*) impacted the expression of DEG for the ALA-H group (Appendix A; Table 3). The DnaJ heat shock protein family (Hsp40) member C1 (*DNAJC1*), TSC22 domain family member 2 (*TSC22D2*), and zinc finger protein 800 (*ZNF800*) were also ranked, although they were not listed within any significant GO:Terms (Appendix A; Table 3). In contrast, the TF forkhead box S1 (*FOXS1*), upstream binding protein 1 (*UBP1*), and nuclear receptor subfamily 2 group C member 1 (*NR2C1*)—“gene expression” (GO:0010467); GDNF inducible zinc finger protein 1 (*GZF1*)—“DNA binding” (GO:0003677); and the zinc finger protein FOG family member 1 (*ZFPM1*) were indicated with the lowest values for RIF1 and RIF2 (Appendix A; Table 3).

### 2.4. Differential Co-Expression Analysis (WGCNA)

The DCO-WGCNA identified 521 genes inserted in six modules significantly correlated (*p* ≤ 0.1) with values higher than |0.4| in the LA-H group (Appendix A). However, only the lightgreen (−0.59, *p* = 0.02, *n* = 123 genes) and skyblue (−0.47, *p* = 0.08, *n* = 126 genes) modules had genes enriched in some GO:Terms. It pointed out seven unique component cellular terms, but no KEGG pathway, biological process, or molecular function (Appendix A). Among these terms, we highlight the “intracellular” (GO:0005622), in which the ATP binding cassette subfamily A member 6 (*ABCA6*) gene—light green module—was inserted.

Under the same parameters, five modules were identified, harboring 1682 co-expressed genes in the LA-L group (Appendix A) enriched in unique 245 GO:Terms, 174 BP, 15 MF, 56 CC (Appendix A), and 9 unique KEGG pathways. We can highlight the SREBF chaperone (*SCAP*) gene in the darkseagreen4 module (−0.46, *p* = 0.08, *n* = 248 genes) enriched in “intracellular” (GO:0005622), as well as the fatty acid desaturase 1 (*FADS1*) and 2 (*FADS2*), fatty acid synthase (*FASN*), and peroxisome proliferator activated gamma receptor (*PPARG*) genes in the purple module (0.45, *p* = 0.10, *n* = 800 genes), enriched in the term “cell differentiation” (GO:0030154) (Appendix A).

For the ALA, a total of 852 genes were listed in six modules (ALA-H, Appendix A) enriched in 111 GO:Terms (69 BP, 5 MF, 37 CC) and 17 KEGG pathways (Appendix A), and 247 genes were inserted in two modules (ALA-L, Appendix A) enriched in 19 GO:Terms (1 BP, 18 CC). Among them, we can emphasize the ELOVL fatty acid elongase 5 (*ELOVL5*)—“single-organism metabolic process” (GO:0044710) in the dark olive-green module (−0.49, *p* = 0.07, *n* = 474 genes, ALA-H) and the acetyl-CoA carboxylase beta (*ACACB*) and 3-hydroxy-3-methylglutaryl-CoA synthase 1 (*HMGCS1*) inserted in the floral-white module (−0.44, *p* = 0.10, *n =* 177 genes, ALA-L) enriched in the term “intracellular” (GO:0005622) (Appendix A).

## 3. Discussion

### 3.1. Differentially Expressed Genes

Among the DEG described for the LA, we can highlight the *ACOX2*, *BDH1*, *ECHS1*, and *IVD* genes, enriched in the term “oxidation-reduction process” (GO:0055114), with higher expression in the LA-H group. The *ACOX2* gene corroborates with results from Lim et al. [19], in which it was also identified in Hanwoo cattle with a high marbling score. This gene was also associated with the *BDH1* gene’s overexpression in Nellore cattle [10], which suggests a relationship with higher concentrations of FA. This relationship can still be justified by the regulatory action of such genes due to the high concentration of unsaturated fatty acids, as can be seen with the oxidative action genes *ECHS1* and *IVD* [20,21], which were also pointed out as differentially expressed between ALA groups.

Particularly in the ALA-H group, the *SDHB* gene, enriched in the BP “oxidation-reduction process” (GO:0055114), was identified as upregulated. This gene encodes a subunit that forms the SDH enzyme, responsible for ATP synthesis steps through the Krebs cycle [22]. Moreover, this same enzyme was related to FA concentration in cattle. According to Jeong et al. [23], the SDH enzyme was overexpressed in samples of *Longissimus dorsi* muscle in Korean cattle after castration, which had a higher rate of IMF compared to non-castrated animals. This fact reiterates the relationship of some genes with the oxidative action and FA profile (i.e., *SDHB* gene), and it suggests a possible regulatory mechanism associated with a high FA concentration.

The *ACSL3* and *FITM1* genes, enriched in the “metabolic process” (GO:0008152), were identified as differentially expressed between the ALA groups. The *ACSL3* gene has regulatory potential in the lipogenesis process due to its relationship with the intracellular uptake of FA, as well as the enzymatic action triggered by its protein product by providing substrate for the triglyceride synthesis and β-oxidation steps [24,25]. This gene was downregulated in the ALA-H group, a finding justified by the high FA rate. On the other hand, the *FITM1* gene was correlated with small deposits of lipids in muscle cells [26] and higher fat accumulation in cell cultures [27]. This fact corroborates our results that showed it to be more expressed in the ALA-H group and suggests a relationship with the high FA concentration.

### 3.2. Differential Co-Expression Analysis (PCIT—DH)

For the DCO analysis aiming to identify DH genes (PCIT-DH), we highlighted the *WDR43*, *ASB5*, *TRAFD1*, and *ERLIN1* genes due to their implications in the LA-H and ALA-H groups. The *WDR43* gene acts in the transcriptional control by binding to RNA polymerase II modulating proteins and promoter regions for mRNA formation [28]. Members of the same gene family were related to the FA profile in Nellore cattle [9,11,29], as well as in the marbling score in Hanwoo cattle [30]. The *ASB5* gene was associated with cell differentiation and proliferation in muscle tissue [31], besides being related to the marbling score in the Nellore [10] and crossbred Wagyu × Hereford cattle [32]. Although the function of the *TRAFD1* gene is not clear, it regulated the DEG expression for IMF content in Iberian swine [33]. The appointment of these genes as DH in the ALA-H and LA-H groups is justified not only by their potential function by regulating gene expression but also because of their impact on lipid metabolism. A similar explanation can be attributed to the *ERLIN1* gene, enriched in the term “regulation of gene expression” (GO:0010468), due to its relationship with the ER lipid raft associated 2 (*ERLIN2*), *SCAP*, and insulin induced gene 1 (*INSIG1*) and 2 (*INSIG2*) for the modulation of the *SREBF1* gene, and consequently, to cholesterol homeostasis establishment [34]. This fact is also related to fat deposition, as observed by Huber et al. [35], when indicating the *ERLIN1* gene as responsible for triglyceride formation in a cell culture study.

The *ORC4* gene—“DNA metabolic process” (GO:0006259) and *NAA15*—“positive regulation of gene expression” (GO:0010628) were highlighted as some of the most expressive DH in the LA-H and ALA-H groups, respectively. The protein encoded by the *ORC4* gene belongs to a protein complex associated with DNA replication [36]. Complementarily, the *NAA15* gene was associated with cell proliferation in tumor cells [37], indicating a possible relationship between these genes and the mechanisms underlying cell proliferation and differentiation in the high groups due to the high cell activity in fat synthesis and storage.

Although no common DH genes were identified between the LA-L and ALA-L groups, the *FN1*, *MMP14*, *FSTL1*, *LRP1*, and *DAB2* genes can modulate the gene expression in the LA-L group. Among them, *FN1* and *MMP14* are closely related to extracellular matrix formation, a fundamental step for cell proliferation and fat deposit formation in muscle tissue [32,38]. The *FN1* gene was overexpressed in pre-adipocytes in cell culture [39], besides being differentially expressed in Nellore cattle with contrasting ribeye muscle areas [40]. The *FSTL1* gene, enriched in the term “response to starvation” (GO:0042594), acts by promoting adipogenesis and fat accumulation in pre-adipocytes and adipocytes differentiated in cell culture, as well as being upregulated in human non-obese patients [41]. The relationship of these genes with the LA-L group can be justified by the low FA concentration, likewise hypothesizing that the animals in this group are later than those in the high group. The *DAB2* gene—“cellular response to lipid” (GO:0071396) and *LRP1*—“positive regulation of gene expression” (GO:0010628) are related to fat deposition in tissues since they act intimately in cholesterol homeostasis [42,43]. According to Tao et al. [44], the *DAB2* gene acts in pre-adipocyte differentiation stages, which is shared by the *LRP1* gene, as presented by Masson et al. [45] in a cell culture study. This association with the early stages of adipocyte formation suggests that the animals in the LA-L group are in prior stages to those in the high group.

Finally, the *DEK* and *KMT2E* genes, enriched in the term “gene expression” (GO:0010467), were identified as the DH genes in the ALA-L group. Although the *DEK* gene is related to tumor processes in humans, it is involved in stages of cell differentiation [46], and complementarity to the *KMT2E* gene, which acts on cell proliferation [47], allows us to reinforce the possibility that ALA-L animals are less precocious. The identification of the *PRPF38B*, *SEC62*, and *USP8* genes as major DH in the ALA-L group does not indicate a direct association with the FA phenotype due to functions related to the protein complex involved in general transcriptional processes [48,49,50].

### 3.3. Differential Co-Expression Analysis (PCIT—RIF)

Common to both FA, the TF *NFIA* and *ZNF473* displayed the highest positive RIF1 and RIF2 values, respectively. The *NFIA* gene was related to the adipogenic processes and cell differentiation since it could regulate the expression of genes such as *PPARG* in non-differentiated adipocytes of cell cultures [51] and in mice brown fat [52]. The transcription factor *ZNF473* was identified as one of the potential regulators of feed efficiency in pigs [53]. Further, this TF was related to energy and lipid metabolism in genomic-wide association and gene co-expression studies in the Nellore cattle [54,55]. These facts corroborate with our results, indicating a relationship of these TF with the EFA metabolism. Moreover, the LA and ALA also shared the TF *HINFP* with lower RIF2 scores. The TF *HINFP* can regulate lipid metabolism due to its interaction with the SREBP2 protein, which regulates cholesterol levels and lipid deposition [56]. In this way, the TF *HINFP* suggests a negative regulatory action under FA biosynthesis.

Similarly, among the TF ranked with the highest RIF1 and RIF2 scores for the LA, we highlight *NFYA* and *NFYB*, which modulate the *FASN* expression in mouse adipocytes, and interact with the SREBP1 protein for FA synthesis and cholesterol homeostasis [57]. Moreover, the identified TF *NFAT5* and *SLC2A4RG* were related to adipogenic processes linked to glucose uptake and cell differentiation [58,59]. This last trait was also linked to the TF *BACH1* in a study carried out by Nishizuka et al. [60] in cell cultures, which indicated it as a potential regulator in adipocyte differentiation. These studies corroborate the potential relationship of these TF with DEG expression and, consequently, fat deposition and the EFA profile, especially for the LA. Moreover, the TF *ZNF134*, *ZNF584*, and *ZBTB43* in RIF1, and *WIZ* (RIF2), also were highlighted. Although these genes have an unclear relationship with lipid metabolism, the zinc finger family members act in the gene expression of several metabolic processes [61], and therefore, these TF can be crucial in regulating DEG expression.

When considering the RIF analysis for the ALA, the TF with the highest RIF1 (*MITF*) and RIF2 (*ATF6* and *HIVEP2*) scores were enriched in the term “gene expression” (GO:0010467). The TF *MITF* indication may have been due to its relationship with the Stearoyl-CoA desaturase (*SCD*) gene, which acts in unsaturated fatty acid synthesis. According to Vivas-García et al. [62], the FT *MITF* promoted higher expression of the *SCD* gene and, consequently, a higher concentration of unsaturated fatty acids in tumor cells. The TF *ATF6* and *HIVEP2* share a similar potential to regulate adipogenic processes since both interact with the *PPARG* gene. According to Lowe et al. [63], the *ATF6* gene knockout significantly reduced the *PPARG* expression in cell culture. The TF *HIVEP2*, also known as *Schnurri-2*, acts as a co-activator in the expression of the *PPARG* gene, and, in knockout situations, restricted the *PPARG* expression during adipocyte differentiation [64]. These relationships reinforce the regulatory potential of these TF under DEG expression in the ALA-H group, and consequently, the EFA profile.

Furthermore, with the ability to impact lipid deposition, TF with higher RIF1 values (ALA, *RORA*, and *NFX1*) were described enriched in the term “gene expression” (GO:0010467). Such impact may be due to the interrelation between these TF and lipogenesis-promoting genes. According to Lau et al. [65], the TF *RORA* has regulatory potential in lipid homeostasis, given that, in knockout situations, this TF induced the lower expression of genes such as *FASN* and *SCD*, crucial for FA synthesis. The FT *NFX1* encodes a protein with the same name that binds to the NF-Y protein, which regulates the *FASN* expression, and, consequently, impacts FA synthesis [57,66]. A relationship was also observed by Fernandes Júnior et al. [67] when pointing to the *NFX1* gene as a candidate gene to explain a part of the genetic variance for backfat thickness in Nellore cattle.

Otherwise, the TF *DNAJC1* (RIF1), *TSC22D2* (RIF2), and *ZNF800* (RIF2) for the ALA act mainly in cell growth control. The *DNAJC1* and *TSC22D2* genes were related to a large number of genes to suppress cell proliferation in tumor processes and adverse conditions [68,69]. The *ZNF800* gene has a similar function, in which it can inhibit cell proliferation by blocking the AKT/mTOR pathway [70]. These TF with higher positive scores suggest a potential regulatory action to control cell proliferation in animals in the ALA-H group.

With lower RIF1 scores for the ALA, TF such as the *FOXS1* and *UBP1*—“gene expression” (GO:0010467) and *GZF1*—“DNA binding” (GO:0003677) were identified. The FT *FOXS1* is associated with several cell proliferation steps [71], and its overexpression inhibited tumor cell proliferation. The FT *GZF1* is also involved in the cell proliferation process; however, it was related to cell proliferation promotion in cell culture [72]. The FT *UBP1* acts in angiogenesis by stimulating neovascularization, an important step in pre-adipocytes to promote growth and differentiation in the cellular as well as extracellular matrix [73,74]. These TF allow us to hypothesize that animals in the ALA-L group are more delayed than those in the ALA-H, indicating early stages of adipocyte formation, and consequently, less lipid deposition.

With low RIF1 scores for the ALA, the TF *ZFPM1* and *NR2C1* are worth highlighting. The FT *ZFPM1* (alias name: *FOG1*) acts as a co-regulator in early adipocyte differentiation stages since it was overexpressed in pre-adipocytes and showed lower expression in mature adipocytes in cell culture [75]. Moreover, *ZFPM1* was pointed out as a DH gene in a low genomic value (GEBV) group for IMF in Nellore cattle [76]. Further related to fat metabolism, the *NR2C1* gene encodes the TR2 protein, which, when interacting with the TR4 protein product, is related to pathways opposite to those presented by the *PPARG* gene, and suppresses the expression of Retinoid X receptors [77]. These findings corroborate with our results, by showing that these TF are related to a lower FA concentration.

### 3.4. Differential Co-Expression Analysis (WGCNA)

The WGCNA methodology employs a different algorithm from that used by the PCIT regarding the gene networks’ formation as well as the gene clustering, complementing each other. This complementarity allowed us to identify other potential DCO genes to improve the understanding of the mechanisms underlying the phenotypes.

The DCO analysis through the WGCNA package identified gene groups important to FA profile development. In the LA-H group, only two modules had GO:Terms enriched, but we highlight the lightgreen module (−0.59, *p* = 0.02), in which the *ABCA6* gene was inserted and pointed out in the component cellular “intracellular” (GO:0005622). According to Wenzel et al. [78], the *ABCA6* gene is related to cholesterol uptake, which is closely linked to the *SREBF1* gene, and its target gene functions in cholesterol homeostasis, impacting the EFA profile [34]. This fact justifies the presence of *ABCA6* in a significant module correlated to LA concentration. For the LA-L group, the *SCAP* gene, enriched in “intracellular” (GO:0005622) and clustered in the darkseagreen4 module (−0.46, *p* = 0.08), also interacts with the *SREBF1* gene, activating it [34]. The protein complex SCAP:SREBP1 promotes, in low-cholesterol conditions, higher synthesis of FA and cholesterol [34], reinforcing the relationship with the EFA profile, justifying the *SCAP* co-expression in a module negatively correlated to the LA-L group. Notwithstanding, the purple module (0.45, *p* = 0.10) pointed out clustered hub genes for the FA phenotype, i.e., *PPARG*, *FASN*, *FADS1*, and *FADS2*, enriched in “cell differentiation” (GO: 0030154). The *PPARG* and the *FASN* genes act in crucial cell proliferation and differentiation stages in adipocytes [79,80,81]. The *PPARG* impacted the lipid deposition in the Hanwoo cattle since it was overexpressed in high marbling score groups [19]. Similarly, the *FASN* gene was correlated (0.44) with IMF (*Longissimus dorsi*) in Korean cattle [82], and it was overexpressed in Iberian pigs with high IMF content [33]. Further, it explained part of the genetic variance for the FA profile in Angus and crossbred cattle musculature [83]. The *FADS1* and *FADS2* genes act in the FA desaturation process. According to Ralston et al. [84], in adipocyte cell cultures, the highest LA concentration triggered the higher expression of *FADS1* and *FADS2*. Moreover, SNP-type mutations in the *FADS2* gene were associated with higher marbling in the Holstein breed [85]. These studies corroborate our results that indicate a positive correlation of the purple module with the LA concentration.

Regarding the ALA, gene groups related to the FA synthesis were identified. The dark olive-green module (−0.49, *p* = 0.07, ALA-H) encompassing the *ELOVL5* gene—“single-organism metabolic process” (GO:0044710) and the floral white module (−0.44, *p =* 0.10, ALA-L) harboring the *ACACB* and *HMGCS1* genes—“intracellular” (GO:0005622) should be underscored. The *ELOVL5* gene acts in stages of unsaturated fatty acid elongation (i.e., C18–20 chains), especially ALA and eicosatrienoic acid (C20:3 n3) [86]. This result corroborates the negative correlation of the dark olive-green module with the ALA concentration. On the other hand, the *ACACB* and *HMGCS1* genes act in the FA oxidation stages [87]. The *ACACB* gene was related to lower lipid deposition in cattle since it was identified by Zhang et al. [47] as downregulated in castrated animal groups, which showed high lipid deposition, compared to non-castrated animals. The *HMGCS1* gene, besides participating in the formation of ketone bodies, is also involved in cholesterol homeostasis and FA uptake together with the same family member, 3-hydroxy-3-methylglutaryl-coenzyme reductase (*HMGCR*) [34,88]. These results corroborate the module’s correlation with the ALA concentration and suggest that the *ACACB* and *HMGCS1* genes’ co-expression is linked to a lower FA concentration.

In summary, the combination of two complementary differential co-expression methodologies, WGCNA and PCIT, with the differentially expressed genes analysis, allowed us to identify hub genes already described in the literature. Furthermore, unexplored genes and transcription factors able to model the EFA profile in zebu meat cattle were identified, revealing regulatory mechanisms linked to lipid and energy metabolism, as well as cell proliferation and differentiation. Although more studies are needed, these findings show the possible genetic potential in zebu breeds raised in tropical climates to improve meat quality and achieve a healthier EFA profile in meat.

## 4. Materials and Methods

### 4.1. Animals and Sampling

We were approved to carry out procedures involving animals by the Animal Use Ethics Committee of the Faculty of Agricultural and Veterinary Sciences (FCAV), Unesp, Jaboticabal/SP (certificate number 18340/16). We used a total of 44 young Nellore bulls, the progeny of six sires, belonging to the Capivara farm (São Paulo state, Brazil), which participated in the Nelore Qualitas breeding program. On this farm, the mating season occurred two times, between the months of February and April, and from mid-November to January. The female animals at 10 to 14 months of age were submitted for three months of the breeding season. Then, the heifers were evaluated by rectal palpation ~60 days after finishing the breeding season to confirm pregnancy. The females who did not get pregnant were exposed again to a breeding season at two years old.

During the raise period, the nutritional management of the animals used in this study was based on grazing conditions using *Brachiaria* sp. and *Panicum* sp. forages; moreover, during the dry season, they were offered creep feeding and supplementation. Then, after yearling, the breeding animals were selected, and the remaining were kept in a feedlot for at least 90 days with a diet based on whole-plant silage, sorghum grain, and soybean meal, with a concentrate/roughage ratio of 70/30. All animals were managed in the same lot from birth to the finishing phase.

Next, on the same day, all animals were slaughtered in commercial slaughterhouses with approximately 550 kg of live weight and an average age of 24 months. At the time of the slaughter, the *Longissimus thoracis* (LT) muscle samples—between the 12th and 13th ribs from the left half carcasses—were collected, properly identified, and subjected to liquid nitrogen. Subsequently, they were stored at −80 °C until further RNA-seq analysis. At the deboning stage, after 48 h of carcass cooling (0–2 °C), in the same place, LT samples (2.5 cm thick) were collected to measure the intramuscular fatty acid profile and content.

### 4.2. Lipid Extraction and Quantification

The lipid fraction quantification was performed at the Animal Product Technology Laboratory in the Technology Department of FCAV/UNESP (Jaboticabal, São Paulo, Brazil) according to Bligh and Dyer [89]. Approximately 3 g of raw and ground samples of *Longissimus thoracis* muscle was subjected to lipid extraction, in which, after being transferred to a 250 mL flask, chloroform (10 mL), methanol (20 mL), and distilled water (8 mL) were added. After this, the solutions were homogenized and centrifuged for 30 min on a horizontal shaker table (HITACHI High-Speed Micro Centrifuge model CF16RN himac). Then, an additional 10 mL of chloroform was added together with 10 mL of an aqueous sodium sulfate solution (1.5%) and agitated for 2 min. This solution, in a 50 mL falcon tube, was centrifuged (1000× *g*-force) at room temperature. The supernatant was discarded and the rest was filtered into measuring cylinders (25 mL) through a paper filter in order to separate the extracted lipid solution. From this last solution, 5 mL was transferred to a 50 mL beaker, dried in an oven, and cooled in a desiccator for at least 24 h. Afterwards, the beaker with the solution was placed in an oven at 110 °C for the evaporation of the solvent, cooled in a desiccator (O/N), and, at the end, weighed. The differences between the initial and final weight of the beaker were used to determine the lipid concentration of the samples.

### 4.3. Fatty Acid Profile Identification

The FA profile was determined for each sample according to Folch et al. [90]. Approximately 100 g of the ground muscle samples was used. The lipids were extracted and isolated by homogenizing a chloroform and methanol solution in 2:1 ratio and subsequent addition of a 1.5% NaCl solution. After this, as proposed by Kramer et al. [91], the methylation process of the isolated lipids was carried out, and this resulted in the formation of methyl esters; from these FA profile compositions, it was possible to perform quantification using a gas chromatographer (GC-2010 Plus—Shimadzu AOC 20i auto-injector) with an SP-2560 capillary column (100 m × 0.25 mm in diameter with 0.02 mm thickness, Supelco, Bellefonte, PA, USA). With an initial temperature of 70 °C, under continuous heating of 13 °C per minute, an intermediate temperature of 175 °C was established and, when reached, was maintained for 27 min. Thereafter, at a heating speed of 4 °C per minute, the temperature was increased to 215 °C and maintained for 31 min.

The FA identification was performed by comparing the retention time of the methyl esters of the samples with standards C4-C24 (F.A.M.E mix Sigma^®^), vaccenic acid C18:1 trans-11 (V038-1G, Sigma^®^), C18:2 trans-10 cis-12 (UC-61 M 100 mg), CLA e C18:2 cis-9, trans-11 (UC-60 M 100 mg), and tricosanoic acid (Sigma^®^). Then, using the GS solution 2.42 software, the FA were quantified, by normalizing the area under the methyl esters curve, and expressed as a percentage of the total FA methyl ester. This step was performed at the Meat Science Laboratory (LCC) in the Animal Nutrition and Production Department at FMVZ/USP. Among the quantified FA, we selected the fatty acids linoleic (C18:2 ω-6) and alpha-linolenic (C18:3 ω-3) due their importance for human health.

### 4.4. RNA Extraction

The extraction of total RNA from 44 bovine LT muscle samples (~100 mg) was performed using the extraction protocol via TRIzol^®^ (Life Technologies, Carlsbad, CA, USA). All the samples used in this study obtained RNA integrity number (RIN) ≥ 8, measured by the Agilent equipment 2100 Bioanalyzer^®^ (Agilent, Santa Clara, CA, USA), and were used. The preparation of the cDNA libraries followed the TruSeq^®^ RNA Sample Preparation v2 protocol (Illumina, San Diego, CA, USA) and their quantifications were performed using the KAPA Library Quantification Kit^®^ (KAPA Biosystems, Foster City, CA, USA). Subsequently, barcode sequences were added for individual identification of the samples, which were subjected to sequencing. The method used for sequencing was the paired-end, which produces fragments of 2 × 100 bp. The HiSeq 2500 equipment (Illumina, San Diego, CA, USA) and the TruSeq PE Cluster kit v3-cBot-HS and TruSeq SBS v3-HS kit (Illumina, San Diego, CA, USA) were used for its realization. The sequencing analysis was performed at the Genome Center at ESALQ/USP (Piracicaba, São Paulo, Brazil).

### 4.5. Read Alignment and Gene Count

HiSeq platform raw sequencing data were converted into FASTQ files and separated into individual libraries using the Casava v.1.8.2 software (Illumina, San Diego, CA, USA). These files were submitted to the software FastQC v. 0.11.9 [https://www.bioinformatics.babraham.ac.uk/projects/fastqc/ accessed on 26 November 2019] and Trimmomatic v. 0.32 [92] to perform quality control, and to identify and remove PCR adapters and primers, as well as low-quality and/or small sequences (<36 bp). The reads were then aligned to the *Bos taurus taurus* genome assembly ARS-UCD1.2 using the STAR v.2.7.0 program [93]. Subsequently, from the SAMtools v1.09 software [94], a second quality control procedure was performed to remove low-quality alignments, secondary alignments, and PCR duplicates. Afterwards, the gene counts were performed by the HTSeq Python package [95], and they were used for subsequent gene expression analyses (DEG and DCO).

Moreover, we evaluated gene expression records on a principal component analysis and hierarchical clustering. However, it was not able to identify a clear group (Figure 1; Appendix A).

### 4.6. Differentially Expressed Genes Analysis

For the differentially expressed analysis (DEG), animals were grouped based on their concentrations for each selected FA (linoleic, LA—C18:2 n6 and alpha-linolenic, ALA—C18:3 n3). Two groups were formed by considering those samples with the 15 highest (high—H) values/phenotypes and the 15 lowest (low—L), totaling four main groups: LA-H, LA-L, ALA-H, and ALA-L. A significant (*p* < 0.0001) difference was observed, by *t* test [96], between the means of the highest and lowest FA concentration groups (Table 4). The DEG analysis was performed through the R package DESeq2 [96] and was used in two EFA-specific groups, where each group had 30 samples (15 high and 15 low). The input gene set was filtered, and genes that had less than one count per million (cpm) in 90% of the samples were filtered out.

### 4.7. Gene Co-Expresion Analysis: Partial Correlation and Information Theory (PCIT)

For the differential co-expression analyses (DCO), the partial correlation and information theory (PCIT) algorithm implemented in R software [16,97] was used (DCO-PCIT). The quality control based on filter genes with less than 1 cpm resulted in 10,739 genes from 27,270 genes. Next, the raw count data were normalized (cpm), and their variance was stabilized by the *variance stabilizing transformation* function on DESeq2 [96]. The correlation matrices for the gene expression values were performed separately for each of the four FA groups evaluated (LA-H, LA-L, ALA-H, and ALA-L) using the PCIT package [97]. From these matrices, only significant correlations ≥|0.9| were used to perform the differential hubbing (DH) analysis. The measurement of DH values was based on the difference in the quantity of significant correlations ≥|0.9| of each gene between the contrasting groups. Genes were then ranked based on the top five DH values (highest and lowest) for each FA.

The regulatory impact factor (RIF) scores were calculated as described by Reverter et al. [98] and they are displayed as *z*-scores. These scores reflect the co-expression of each transcription factor (TF), previously informed, and its potential target genes, i.e., differentially expressed genes identified among contrasting EFA groups. There are two different ways to calculate the RIF measurements. In the first one (RIF1), the RIF score is strongly influenced by large differences in the correlation values between TF and DEG between the divergent groups. In the second one (RIF2), the RIF score is affected mainly by the magnitude of the potential target genes’ (DEG) expression induced by TF expression. The positive values for both scores indicate a better connection with the groups with high concentrations of the assessed fatty acid and, therefore, the opposite situation reveals a relationship with the low groups. For the TF selection, 740 TF were considered, which were expressed among the FA groups and were deposited in the Animal Transcription Factor Database (AnimalTFDB) v.3.0 [99]. Only the TF which had their scores with ±2.58 standard deviation (99%) in relation to the mean were selected for functional and network analysis.

### 4.8. Gene Co-Expresion Analysis: Weighted Gene Co-Expression Network Analysis (WGCNA)

For the differential co-expression analysis (DCO), the weighted gene co-expression network analysis (WGCNA) package implemented in R software [15] was used (DCO-WGCNA). The two specific EFA groups (LA and ALA) that included the highest (*n* = 15) and lowest (*n* = 15) samples were analyzed. As in PCIT analyses, the raw gene counts were normalized (cpm) and had their variance stabilized (*variance stabilizing transformation*) as well as filtered. These groups were analyzed individually, and the connectivity values were calculated among all the genes in the network. The connectivity measure represents the degree of connection between each pair of genes in the network, and it is obtained through the sum of Pearson’s correlations raised to the soft power threshold (β). It should be noted that β is individual for each analysis, and it should approach the criterion of scale-free topology (R^2^ ≥ 0.8) [100]. The values of β = 9 and 8 were used for LA-H and ALA-H and β = 13 and 10 for LA-L and ALA-L, respectively. For the gene clustering and module formation, the unsigned step-by-step method was used, implemented in the hierarchical clustering algorithm (dynamic tree-cutting) package [15]. A minimum size of 30 genes per module and a dissimilarity of >0.20 among the modules were applied [15,101].

Each module has an eigengene measurement (ME), which corresponds to the representation of the first main component of the expression profile of the entire module [15]. This measure was correlated with the evaluated traits in the study, and modules that showed correlations of ≥|0.4| as significant (*p* ≤ 0.1) were selected for further analysis.

### 4.9. Functional Enrichment

The Database for Annotation, Visualization, and Integrated Discovery (DAVID) v6.8 tool [102,103] was used to identify overrepresented gene ontology (GO) terms and KEGG pathways using the list of genes from the differential analyses and the *Bos taurus* annotation file as a background. The list of genes identified in the DEG and both DCO (PCIT and WGCNA) analyses was subjected separately to functional enrichment. The *p*-values were adjusted by False Discovery Rate (FDR) [104], and significant GO:Terms—biological process (BP), molecular function (MF), and component cellular (CC)—and KEGG pathways were considered when *p*-adj ≤ 0.05.

## 5. Conclusions

The DEG and DCO analyses combined allowed us to point out potential unexplored genes/transcription factors and biological processes underlying the EFA (LA and ALA) profile in the *Longissimus thoracis* muscle of Nellore cattle finished in the feedlot. Among them, we can highlight the differentially expressed genes *ECHS1* and *IVD*, as well as the hubbing genes *ASB5* and *ERLIN1,* and the TF *NFIA*, since these genes displayed an outstanding capacity to strongly impact both EFA, linoleic and alpha-linolenic acids. Furthermore, the TF *NFYA*, *NFYB*, *FASN*, and *PPARG* and *FADS2* genes associated with the LA, as well as the TF *RORA* and *ELOVL5* gene with ALA, were able to directly impact the phenotypes.

Our findings contribute to the pinpointing of the potential biomarkers for this complex, arduous, and late measuring trait. Moreover, our results provide information that would help us to better understand the genetic and physiological mechanisms related to the synthesis of essential fatty acids in cattle. Furthermore, they can support -omics studies, especially genomic-wide association studies and genomic selection, since this information could be used to weight some makers located on transcriptional factors or nearby regulatory genes, which can regulate gene expression and related pathways. Therefore, this also could assist in the early selection of animals with a healthier essential fatty acid profile.

## Figures and Tables

**Figure 1 metabolites-12-00471-f001:**
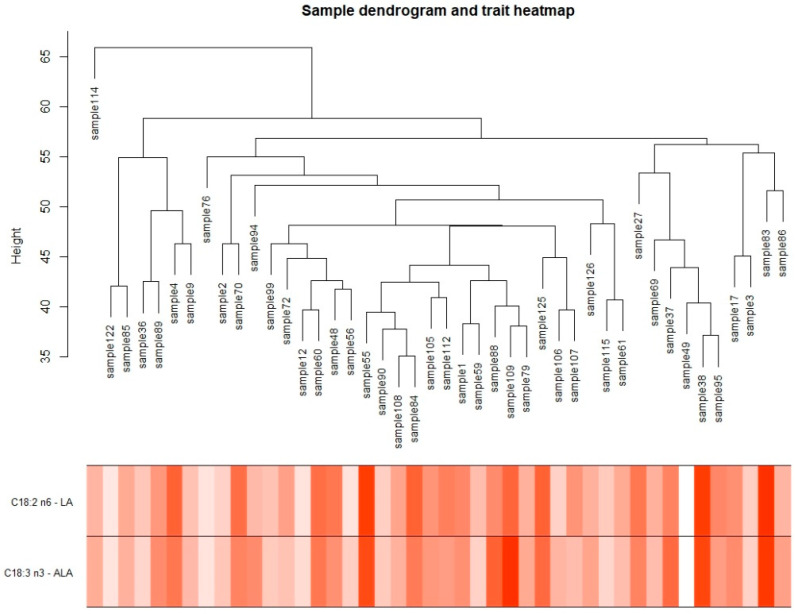
Sample dendrogram and trait heatmap for each essential fatty acid (linoleic and α-linolenic acid). The range of colors represents the intensity of the trait based on its phenotype.

**Table 1 metabolites-12-00471-t001:** List of extreme differential hubbing genes for the linoleic acid (LA—C18:2) and alpha-linolenic (ALA—C18:3) fatty acids.

Groups	Gene Symbol	DH	Groups	Gene Symbol	DH
	*WDR43*	68		*ASB5*	40
	*ASB5*	63		*TRAFD1*	35
LA-H	*ORC4*	60	ALA-H	*WDR43*	31
	*ERLIN1*	57		*NAA15*	30
	*TRAFD1*	50		*ERLIN1*	30

	*FSTL1*	−70		*DEK*	−91
	*FN1*	−66		*PRPF38B*	−89
LA-L	*DAB2*	−63	ALA-L	*KMT2E*	−77
	*LRP1*	−62		*USP8*	−77
	*MMP14*	−60		*SEC62*	−75

LA-H: Samples with the 15 highest values/phenotypes for the linoleic acid; LA-L: Samples with the 15 lowest values/phenotypes for the linoleic acid; ALA-H: Samples with the 15 highest values/phenotypes for the alpha-linoleic acid; ALA-L: Samples with the 15 lowest values/phenotypes for the alpha-linoleic acid.

**Table 2 metabolites-12-00471-t002:** List of genes with the highest and lowest regulatory impact factor (RIF) scores (RIF1 and RIF2) between the contrasting linoleic acid (LA—C18:2) groups.

Groups	Gene Symbol	RIF1	RIF2
**RIF1 (+)**	*ZNF134*	4.91	−1.01
	*NFIA*	4.50	−0.39
	*NFYA*	4.07	−0.84
	*NFAT5*	4.02	0.78
	*ZNF584*	3.54	0.07
**RIF2 (+)**	*ZBTB43*	−0.46	2.95
	*ZNF473*	−0.09	2.83
	*SLC2A4RG*	−0.34	2.77
	*BACH1*	−0.17	2.73
	*NFYB*	2.87	2.68
**RIF2 (−)**	*HINFP*	1.02	−2.70
	*WIZ*	−0.63	−2.65

**Table 3 metabolites-12-00471-t003:** List of genes with the highest and lowest regulatory impact factor (RIF) scores (RIF1 and RIF2) between the contrasting alpha-linoleic acid (ALA—C18:3) groups.

Groups	Gene Symbol	RIF1	RIF2
**RIF1 (+)**	*NFIA*	7.43	−1.05
	*DNAJC1*	5.11	0.67
	*RORA*	4.94	0.46
	*NFX1*	4.80	2.27
	*MITF*	4.75	1.14
**RIF1 (−)**	*FOXS1*	−3.18	0.15
	*UBP1*	−2.81	0.14
	*GZF1*	−2.68	0.73
**RIF2 (+)**	*TSC22D2*	−0.46	2.95
	*ZNF800*	−0.09	2.83
	*ATF6*	−0.34	2.75
	*HIVEP2*	−0.17	2.73
	*ZNF473*	2.87	2.68
**RIF2 (−)**	*NR2C1*	0.16	−3.57
	*HINFP*	−0.32	−2.68
	*ZFPM1*	0.19	−2.65

**Table 4 metabolites-12-00471-t004:** Descriptive statistics of the phenotypic data evaluated in the two contrasting groups and means comparison.

Fatty Acid	Low Group (*n* = 15)				High Group (*n* = 15)				
	Min	Max	Mean	SD	Min	Max	Mean	SD	*p*-Value
**LA**	2.47	5.79	4.57	0.89	8.00	11.83	9.39	1.19	*
**ALA**	0.23	0.55	0.46	0.08	0.78	1.21	0.94	0.14	*

* *p*-value < 0.0001.

## Data Availability

The RNA-seq data were submitted to the Sequence Read Archive (SRA) database and are available under accession number: PRJNA780472.

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
