# Peer review of "Transcriptome Profile Reveals Genetic and Metabolic Mechanisms Related to Essential Fatty Acid Content of Intramuscular Longissimus thoracis in Nellore Cattle"

_metabolites, 2022, doi:10.3390/metabo12050471_

Round 1

Reviewer 1 Report

The study by Schettini et. al. uses state of the art technology to like gene expression and coexpression patterns to EFA content in zebu breeds. Understanding how the transcriptional phenotypes of different cattle breeds affect desirable traits in meat could be useful for positive selection, as well as to advertise or create a ‘trade mark’ in a world with rising concerns about carbon footprint and health issues associated with red meat consumption. The topic of the study is highly relevant for beef producing countries. This reviewer enthusiasm, however, was dumped by several technical and methodological concerns described below.

Major comments:

Authors must provide strong supportive evidence, and multiple references, showing that RNAseq studies on tissues extracted 48hs post-mortem are meaningful. Experimental data showing RNAseq expression profiles in fresh muscle, and at 1hr, 24hs and 48hs post-mortem are necessary to validate this approach.

Authors should clarify their methods, as it reads now, it looks like the input for DESeq2 was cpm, while this program was designed to use raw counts. Also unclear why p-values were further adjusted as DESeq already informs nominal and adjusted p values.

Also in DESeq2, the group’s selection seem rather arbitrary. An alternative could be to compare one trait at the time, while controlling for other in the design formula. For instance to uncover genes associated with the concentration of linoleic acid, divide the whole cohort into linoleic acid tertiles, and compare the upper to the lower tertile to have more power. Controlling for other FA in the design formula is recommended.

In order to make comparisons using WGCNA, in particularly intramodular connectivity measurements, the soft power threshold should be the same.  The standard procedure uses all samples to create a WGCNA network which is blind to experimental groups, and then makes pairwise comparison between groups by plotting the module eigenvector of each sample against a paired physiological trait and conducting correlation analyses. In your case, as you have continuous traits, you don’t even need to use groups and can use the traits itself. This will add power to your analysis, and is also a less biased design. I would also suggest to include the weight of the carcasses as a trait, as it will provide you with more data. This last will not be necessary for this manuscript but could be useful for future studies.

Following the identification of differentially coexpressed genes, authors can calculate z-cores of target gene groups as a measure of activation and conduct Kaplan-Meier analyses with the lower vs upper half/tertile of EFA content. These studies usually correlate well with WGCNA and add a new line of evidence.

The manuscript is mostly descriptive. A more critical interpretation of results is needed.

Are there any genes that stick out on all analyses? What does this means?

Minor comments:

Lines 32 to 38 “0. How to Use this template” should be removed.

Line 54: “SNP” was not defined.

Line 411-413: It says twice that samples were frozen in liquid N2.

Line 461: the number of samples that passed the RNAseq QC should be informed.

Lines 485-488: I suggest using WGCNA clustering features, which also plot the traits in a color scale under the tree. This should go on main text.

Reviewer 2 Report

This manuscript uses RNA expression data with differentially expressed genes analysis combined with two co-expression algorithms (WGCNA) and (PCIT) to identify potential regulatory transcription factors and hub gene groups to the EFAs profile in the Longissimus thoracis muscle of Nellore cattle. The experimental design and the use of the number of biological replicates sound convincing. Overall, this is an interesting study. There are some improvements required to be accepted for publication.

1.    I do not mind using supplementary data for publication, especially online publication. However, the supplementary figures should treat like a formal publication such as a clear legend for the reader to understand the meaning of the figure. So please add the legends to all the supplementary figures! 
2.    The sequential number of supplementary figures: Due to the format of the journal, the methods section is at the end of the text. So as the Supplementary Figure 1 and 2. Could you re-arrange the sequential number for the supplementary figures?
3.    The supplementary table 14 was mentioned in the text several times, however, it is not in the supplement fold. 
4.    Line 150: (Supplementary File 11…) What is that?
5.    Supplementary Figures 1 and 2: while green is high-group and red is low-group, what is black?

Reviewer 3 Report

Manuscript ID:  metabolites-1676367

Title: Transcriptome profile reveals genetic and metabolic mechanisms related to essential fatty acid content of intramuscular Longissimus thoracis in Nellore cattle

General comments and judgment

The Authors aimed to identify potential regulatory transcription factors and hub gene groups related to the EFAs profile in the Longissimus thoracis muscle of Nellore cattle finished in feedlot based on two methodologies (WGCNA and PCIT), with several purposes, including to assist in the early selection of animals with a healthier essential fatty acid profile.

The manuscript is well written, in fluent English language: clear, precise, and easy to understand. Also, it offers interesting insight on the need to expand the knowledge about the genetic and physiological mechanisms related to the synthesis of essential fatty acid in cattle and to deepen -omics studies on this topic. I suggest a careful re-reading of the entire manuscript in which sometimes there are spelling errors and broken sentences, but in general the reading is pleasant. However, some point remains to be clarified, as specified below in the “Main comments” section.

Main comments

  • In the Materials and Methods section, I recommend to add a brief description of the breed and the type of breeding on the Capivara farm, to allow a greater understanding of the results.
  • In particular, please add information on farming and feeding conditions: are they kept indoor, outdoor or both? How are they fed? Do they graze on natural pasture?
  • Please add some information also on the slaughtering conditions: are they slaughtered in the same days or time period? And if not, could the different season interfere with the obtained results?

I think all these factors could be properly discussed.

Round 2

Reviewer 1 Report

No further comments.

Author Response

No comments have added to the review.